# Acute vaping exacerbates microbial pneumonia due to calcium (Ca$^{2+}$) dysregulation

Rui Zhang[1,2©], Myles M. Jones[3©], De'Jana Parker[3], Ronna E. Dornsife[2], Nathan Wymer[4], Rob U. Onyenwoke[2,5], Vijay Sivaraman[3]*

**1** Department of Respiratory and Critical Care Medicine, General Hospital of Ningxia Medical University, Yinchuan, Ningxia, People's Republic of China, **2** Biomanufacturing Research Institute and Technology Enterprise (BRITE), North Carolina Central University, Durham, North Carolina, United States of America, **3** Department of Biological and Biomedical Sciences, North Carolina Central University, Durham, North Carolina, United States of America, **4** Department of Chemistry and Biochemistry, North Carolina Central University, Durham, North Carolina, United States of America, **5** Department of Pharmaceutical Sciences, North Carolina Central University, Durham, North Carolina, United States of America

© These authors contributed equally to this work.
* vijay.sivaraman@nccu.edu

**Data Availability Statement:** All relevant data are within the manuscript and its Supporting Information files.

## Abstract

As electronic cigarette (E-cig) use, also known as "vaping", has rapidly increased in popularity, data regarding potential pathologic effects are recently emerging. Recent associations between vaping and lung pathology have led to an increased need to scrutinize E-cigs for adverse health impacts. Our previous work (and others) has associated vaping with Ca$^{2+}$-dependent cytotoxicity in cultured human airway epithelial cells. Herein, we develop a vaped e-liquid pulmonary exposure mouse model to evaluate vaping effects *in vivo*. Using this model, we demonstrate lung pathology through the use of preclinical measures, that is, the lung wet: dry ratio and lung histology/H&E staining. Further, we demonstrate that acute vaping increases macrophage chemotaxis, which was ascertained using flow cytometry-based techniques, and inflammatory cytokine production, via Luminex analysis, through a Ca$^{2+}$-dependent mechanism. This increase in macrophage activation appears to exacerbate pulmonary pathology resulting from microbial infection. Importantly, modulating Ca$^{2+}$ signaling may present a therapeutic direction for treatment against vaping-associated pulmonary inflammation.

## Introduction

Electronic cigarettes, also known as E-cigs, are electronic devices used to heat a liquid to create an aerosol that is inhaled, or "vaped", by the user. The liquids in E-cigs, known as e-liquids, contain various compounds, which often include propylene glycol (PG), vegetable glycerin (VG), nicotine and various flavoring agents [1, 2]. With more than 400 E-cig brands available, E-cigs and e-liquids are available in a wide variety of compositions [3]. This variety of flavors and the availability of customization have made E-cigs extremely popular among middle and

**Funding:** This work was supported in part by a grant from the National Cancer Institute (NCI) NIH 5-U54-CA156733-10, National Minority Health Disparities Institute (NIMHD) NIH-3-U54MD012392 and by funds from NCCU BRITE.

**Competing interests:** The authors have declared that no competing interests exist.

high school students as well as young adults [4–6]. Following the increase in E-cig use and more recently (late 2019 to early 2020) reported cases of E-cig or vaping-associated lung injury (EVALI) [7], the overall health impacts of these new products have been called into question. *In vitro* and *in vivo* studies have demonstrated the detrimental effects of these products on pulmonary health [8–11]. Furthermore, a previous assessment of JUUL e-liquids has illustrated notable signs of cytotoxicity in cultured airway epithelial cells [12]. Studies have also tied together an observable cytotoxicity after e-liquid exposure with increases in cytoplasmic calcium ($Ca^{2+}$) [12–14]. These earlier studies led us to further investigate the cellular effects of e-liquids *in vivo*.

$Ca^{2+}$ concentrations impact almost every cellular process, e.g., protein stability and programmed cell death [15]. In addition, $Ca^{2+}$ is vital to cell signaling and to regulating inflammatory responses [16]. Multiple $Ca^{2+}$ channels and pumps are involved in its tight regulation and include the non-store-operated calcium (SOC) transient receptor potential (TRP), the SOC Orai1 and ER-resident inositol trisphosphate ($IP_3$) channels [15, 17]. Alterations in cellular signaling pathways such as SOC signaling and the activation of downstream SOC kinases such as PKCα lead to alterations in rates of apoptosis and cell proliferation [18].

We previously examined the responses of cultured airway epithelial cells to various e-liquids [12]. Here, we report on a series of experiments further investigating the pulmonary inflammatory response after exposure to "vaped" e-liquid in an established *in vivo* model. Using this exposure model, we questioned first whether acute vaping would recapitulate our previously published *in vitro* findings, and whether vaping might "prime" the immune inflammatory response and result in a worse prognosis after a microbial challenge, that is, vape pre-exposure would result in a worse outcome compared to no pre-exposure. For this study, we utilized our previously published *Klebsiella pneumoniae* mouse model [19] and evaluated for severity of disease using several metrics: gross lung morphology and weight, inflammatory cytokine panels, lung histology/H&E staining and immune cell profiles. At the same time, we also treated with well-described $Ca^{2+}$ receptor antagonists to determine the effects of such compounds, if any, on inflammatory responses.

## Materials and methods

### Materials and cells

*Klebsiella pneumoniae* (BA-1705) was obtained from the American Type Culture Collection (ATCC, Manassas, VA, USA). *K. pneumoniae* cultures were routinely grown for 12–16 h in Brain Heart Infusion (BHI) broth (37°C). Bacterial numbers were calculated using serial dilution plating on BHI agar plates.

Unless otherwise noted, all reagents were purchased from either Thermo Fisher Scientific (Waltham, MA, USA) or Sigma-Aldrich (St. Louis, MO, USA) at the highest level of purity available. The utilized JUUL e-liquids/ "pods" were purchased locally from retailers in Durham, NC, USA from November 1, 2019—July 31, 2020. These products were inventoried and stored at room temperature until used. However, after being vaped using the JUUL E-cig device, the vaped e-liquid was stored at -20°C. The manufacturer's label information stated ingredients include only vegetable glycerin (VG), propylene glycol (PG), nicotine, flavoring and benzoic acid.

### Vape distillate generation and nicotine analysis method

The e-liquid was vaped using a previously described [20] method to produce a vapor distillate. Briefly, e-liquid vapors were produced using prefilled pods and a JUUL E-cig device connected to silicon tubing and to the mouthpiece of the E-cig on one end. The other end was placed in

the lower part of a 50-ml conical tube. The distillate was condensed and collected while suspended above liquid nitrogen inside of a thermal container. The JUUL device was utilized for periods of up to 5 s with at least 10 s between activations to simulate "puffs".

The LC-MS analysis of nicotine within the vaped e-liquid samples was performed similarly as described by Pagano and coworkers [21]. Specifically, nicotine was serially diluted in LC-MS-grade methanol to the following concentrations: 1.0, 0.5, 0.1 and 0.05 mg/ml. Quinoline (45.7 μl) was diluted into 1 ml methanol. The diluted quinolone (10 μl) was added per 1 ml of each sample or standard. The vaping sample was diluted 10-fold in methanol. The samples were submitted to the Mass Spectrometry Laboratory at the Louisiana State University and Agricultural and Mechanical College (LSU) in Baton Rouge, Louisiana.

The samples were analyzed with an Agilent 5977A mass selective detector (MSD) attached to an Agilent 7890B gas chromatograph (GC). The system used a 30m DB5-MS Ultra Inert (Agilent) with a 0.25 mm inner diameter and a 0.25-μm film thickness. DB5 column contains a nonpolar phenyl arylene polymer, virtually equivalent to (5%-phenyl)-methylpolysiloxane. GC Method: Starting temp: 80˚C held for 4 min, increasing at a rate of 6˚/min to 300˚C, and then held for 15 min. The injector was run in splitless mode at a temp of 250˚C. Carrier gas was helium at a constant flow rate of 1.5 ml/min. MS Parameters: Transfer temp = 250˚C, source temp = 230˚C, quad temp = 150˚C, and a solvent delay of 4 min. Nicotine was detected using the 84.00 m/z (83.70–84.70 m/z) fragment while the quinoline was detected with a 129 m/z (128.70–129.70 m/z) signal. The nicotine eluted at 12.6 min on the GC while the quinoline eluted at 10.3 min. The nicotine content was determined to be ~2 mg/ml, which is in contrast to an unvaped e-liquid concentration of 35 mg/ml [22]. Because each JUUL pod (0.7 ml) can produce ~200 "puffs" (juul.com), a "puff" can be calculated to contain ~125 μg of nicotine. However, for our studies, we utilized 10 μl (20 μg nicotine) doses of our vaped e-liquid (~2 mg/ml) to attempt to minimize any potential morbidity.

## Mice and treatments

Mice (C57-BL/6J) were from Jackson Laboratories (Bar Harbor, MA, USA). Young adult (6- to 8-week-old male and female) mice were used for all experiments [23]. After delivery, the mice were allowed to recover from shipping stress for 1 week at the NC Central Univ. Animal Resource Complex, which is accredited by American Association for Accreditation of Laboratory Animal Care. All animal care and use were conducted in accordance with the Guide for the Care and Use of the Laboratory Animals (National Institutes of Health) and approved by the NCCU IACUC. Mice were maintained at 25˚C and 15% relative humidity with alternating 12-h light /dark periods.

Once acclimated, mice were anesthetized using isoflurane and a SomnoSuite low-flow anesthesia system (Kent Scientific Corporation, Torrington, CT, USA). The e-liquid distillate, vehicle control (50:50 (vol/vol) PG/VG) or saline (10 μl) was then delivered dropwise intranasal (IN) using a micropipette, as has been previously described [24, 25], once daily for 3 days to each animal in the appropriate treatment group. On the 4th day, the mice were sedated with an i.p. injection of ketamine (100 mg/kg) and xylazine (50 mg/kg) and infected IN with a sub-lethal dose ($2 \times 10^4$ CFU) of *K. pneumoniae* suspended in 20 μl 1× phosphate buffered saline (PBS), as previously described [19]. Body weight and health conditions was monitored daily per IACUC protocols. Humane endpoints were not used for this study as experimental time points of completion were chosen before any significant body weight change clinical signs of disease can be observed. At time points of experimental completion (24 hpi), mice were humanely euthanized using $CO_2$ asphyxiation and cervical dislocation, as per our accepted

NCCU animal protocol and NCCU Animal Resource Complex housing guidelines and conditions.

## Lung wet/dry ratio

Lungs were immediately removed from euthanized mice (24 hours post infection) and weighed (wet weight). The lung tissue was then dried in an incubator (65°C) for 24 h and reweighed (dry weight). The wet/dry ratio was calculated by dividing the wet weight by the dry weight.

## Sample preparation from lungs

Bronchoalveolar lavage (BAL) fluid was obtained as has been previously described [19]. In brief, mice were euthanized, and a catheter attached to 1-ml syringe was inserted into the trachea. The syringe was then used to deliver 1× PBS, which was gently pipetted up and down 3× to remove fluid. BAL fluid was then directly utilized for flow studies or clarified via centrifugation for Luminex analysis. Alternatively, total lung was dissected out, then homogenized in PBS containing protease inhibitors, clarified via centrifugation and quantified using the Bradford assay.

## Luminex analysis

Cytokine panels were assessed using a 10-plex mouse Luminex kit (cat. #SPRCUS1264, MILLI-PLEX, EMD Millipore, Chicago, IL, USA) and either BAL fluid or clarified whole mouse lung cell lysates.

## Flow cytometry

Cells isolated from BAL were stained with fluorescent-tagged antibodies as per the manufacturers' instructions. CD170 (SiglecF)-FITC, Ly6G-PE, F4/80-AF700, CD11B-APC, CD11C-Superbright 780 and eBioscience Fixable Viability Dye eFluor 450 were purchased from Thermo Fisher Sci (eBiosciences). In brief, antibodies were chosen to sort mouse macrophages and neutrophils, similar to previously published work [26]. Data acquisition was performed using a 3-color CytoFLEX flow cytometer (Beckman Coulter, FL, USA). The CytExpert acquisition software (Beckman Coulter) was used to acquire samples while FCS Express (version 6; DeNovo Software, Pasadena, CA, USA) was then used for the analysis. For each sample, at least 30,000 events were collected.

## Histopathology

At time points of experimental completion, mice were humanely euthanized using an overdose of $CO_2$ or isoflurane. Lungs were inflated with 1 ml of 10% neutral-buffered formalin, removed and then suspended in 10% formalin for 12 h. Lungs were washed once in PBS and then immersed in 70% ethanol. Tissues were then embedded in paraffin, and three 5-μm sections (200 μm apart per lung) were stained using hematoxylin/eosin (H&E) for examination by the NCCU Histopathology Core (directed by Dr. X Chen). Histology was scored for damage associated with acute lung injury (ALI) using a rubric described by the American Thoracic Society. In short, 200X magnified H&E-stained sections were evaluated for A) neutrophils in the alveolar space, B) neutrophils in the interstitial space, C) hyaline membranes, D) proteinaceous debris filling the airspaces and E) alveolar septal thickening, with a score generated using the equation: ((20xA)+ (14xB)+ (7xC)+ (7xD)+ (2xE)/(number of fields x 100) [27].

## Statistics

Statistical analyses were performed using GraphPad Prism (La Jolla, CA, USA) and Microsoft Excel analysis. Appropriate statistical tests (Student's *t*-test, ANOVA) were determined after discussion with NCCU biostatistics faculty. Unless otherwise indicated, the results are shown as the mean ± the standard error of the mean (SEM). A value of $P<0.05$ is considered as statistically significant.

# Results

## "Vaped" e-liquid exposure results in cytotoxicity and a pro-inflammatory response

Our previous study investigated the consequences of exposure to resting, or "unvaped", e-liquids using cultured airway epithelial cells [12]. Those data clearly demonstrated increased pro-inflammatory responses and cytotoxicity after the e-liquid exposures. Herein, we evaluate e-liquids after the "vaping" process. To that end, we developed a protocol utilizing commercially available E-cig devices to aerosolize the e-liquid and then condensed the vapors back into a "vaped" e-liquid. With this new reagent in hand, we recently assessed its toxicity *in vitro* [28]. We also performed H&E lung histology and found that the vehicle- and unvaped e-liquid-treated samples more similar to each other when compared to the vaped-e-liquid-treated tissue, which displayed more marked pathology, that is, alveolar wall thickening and the infiltration of inflammatory cells into the interstitial space (**S1 Fig**).

## Vape pre-exposure exacerbates *K. pneumoniae*-induced lung pathology

We next used our acute vape exposure model and infected animals with a published sub-lethal dose ($2 \times 10^4$ CFU) of *K. pneumoniae*, which is a well described model for hospital acquired infections [29], known to induce early and controlled inflammatory responses post-pulmonary inoculation [30] and, therefore, ideally suited for modelling pulmonary pathology. The mice were then monitored post-infection before euthanizing the animals 24 hpi (**Fig 1A**). Because of the acute nature of infection, we did not expect to observe clinical disease behavior, though hunching and lethargy were observed in the infected animals post-vape exposure. After dissection, the lungs were weighed both wet and after drying and used to calculate the "wet/dry" ratio [31, 32], with an increase in this ratio being a well-described preclinical metric of acute lung injury and reflecting severity of lung inflammation. From these observations, it could be noted that the lungs from the vaped e-liquid pre-exposure followed by bacterial infection mice had significantly higher wet/dry ratios compared to the vaped vehicle pre-exposure followed by bacterial infection group (and bacterial infection alone) (**Fig 1B**). The lung pathology of the *K. pneumoniae*-infected only animals also demonstrated a mild inflammatory influx of neutrophils within the alveoli, though the majority of airways were unimpeded (**Fig 1C**). However, the lungs from animals vaped prior to *K. pneumoniae* infection demonstrated more significant inflammatory influx, characterized by monocytic and neutrophilic cells overwhelming within the alveoli. Acute Lung Injury (ALI) scoring of tissue sections clearly demonstrated significant disease burden within this group (**Fig 1C**). BAL fluid was then evaluated for the presence of inflammatory cytokines by Luminex analysis (**Fig 1D**). As expected, the innate immune cytokines TNFα, IL-6 and Mip-1α were upregulated in the *K. pneumoniae*-infected lungs, with the levels significantly increased within the lungs of the vaped animals. Interestingly, IL-17 levels were also observably activated in the vaped and *K. pneumoniae*-infected animals, suggesting a potential role for other inflammatory cells such as natural killer T cells.

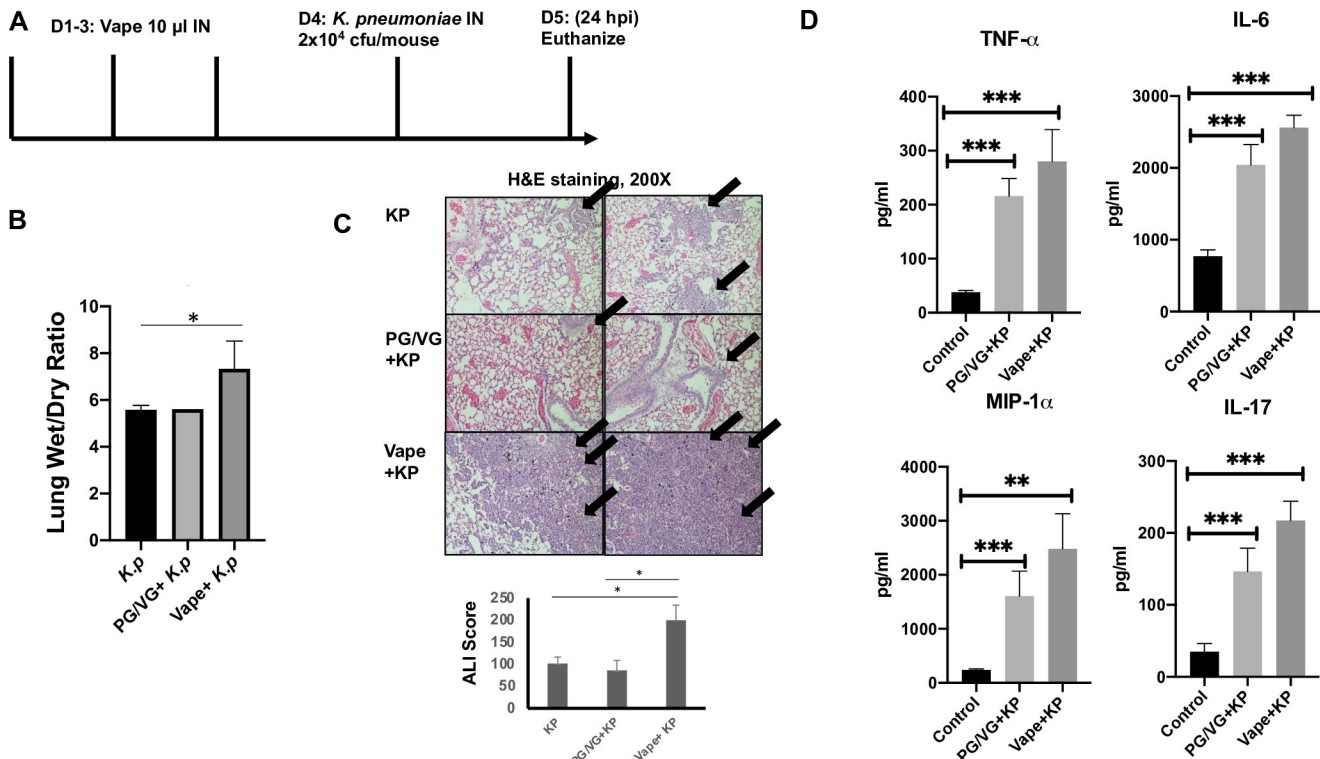

**Fig 1. Vaping exacerbates key clinical metrics of microbial pneumonia.** (A) Study design. (B) Lung wet/dry ratio of mice pre-exposed to either vaped vehicle or e-liquid and then infected with a sub-lethal dose of *Klebsiella pneumoniae.* n = 4 mice per treatment group. (C) H&E staining (200X magnification) of lung tissue isolated from vaped vehicle- or vaped e-liquid- treated mice infected with *K. pneumoniae.* n = 3 mice per group. Arrows indicate alveoli thickening and neutrophil influx. ALI was evaluated and scored to quantify inflammation within tissue sections using an ATS accepted rubric [27]. (D) Inflammatory cytokine analysis of BAL fluid from vaped vehicle- or vaped e-liquid- treated mice infected with *K. pneumoniae* (compared to the mock control) as performed via Luminex array. Symbols and bars represent the mean ± SEM compared to the mock control (* P<0.05, ** P<0.01, *** P<0.001).

## Role of calcium (Ca$^{2+}$) signaling in vape-primed lung pathophysiology

Based upon our previous *in vitro* studies investigating the role of Ca$^{2+}$ in e-liquid-induced cytotoxicity and inflammation [12, 14] and our observations of enhanced acute lung pathology after vaped e-liquid pre-exposure and *K. pneumoniae* infection in comparison to the vaped vehicle followed by *K. pneumoniae* challenge (**Fig 1B and 1C**), we further queried the potential of a mechanistic role for Ca$^{2+}$ signaling in this pathophysiology. To dissect out this mechanism, selective pharmacology was employed to distinguish between the major channels involved in Ca$^{2+}$ influx due to tobacco/E-cig exposure, that is, store-operated calcium entry (SOCE), non-SOCE and the ER-resident inositol trisphosphate (IP$_3$) receptor, which is the receptor regulating calcium release into the cytosol when activated [33–35]. The respective compounds used to selectively target these channels were Pyr6, Pyr10 and 2-APB [36, 37].

For this first study, we followed the treatment regime outlined in **Fig 1A**. However, along with the vape treatments on days 1–3, the mice also received either Pyr6, Pyr10, 2-APB or no drug treatment IN (10 µl of a 500 µM stock) 30 min prior to the vape treatment (**Fig 2A**). The mice were also not infected with *K. pneumoniae*. On day 4, the animals were euthanized, and bronchoalveolar lavage (BAL) fluid was obtained for Luminex (**Fig 2B**) and flow-based assays (**Figs 2C, 2D, S3A and S3B**). From the Luminex data (**Fig 2B**), it was clear that the animals receiving either Pyr6, Pyr10 or 2-APB exhibited significantly lower levels of the pro-inflammatory cytokine IL-6 as compared to the mice receiving only the vape exposure. Additionally,

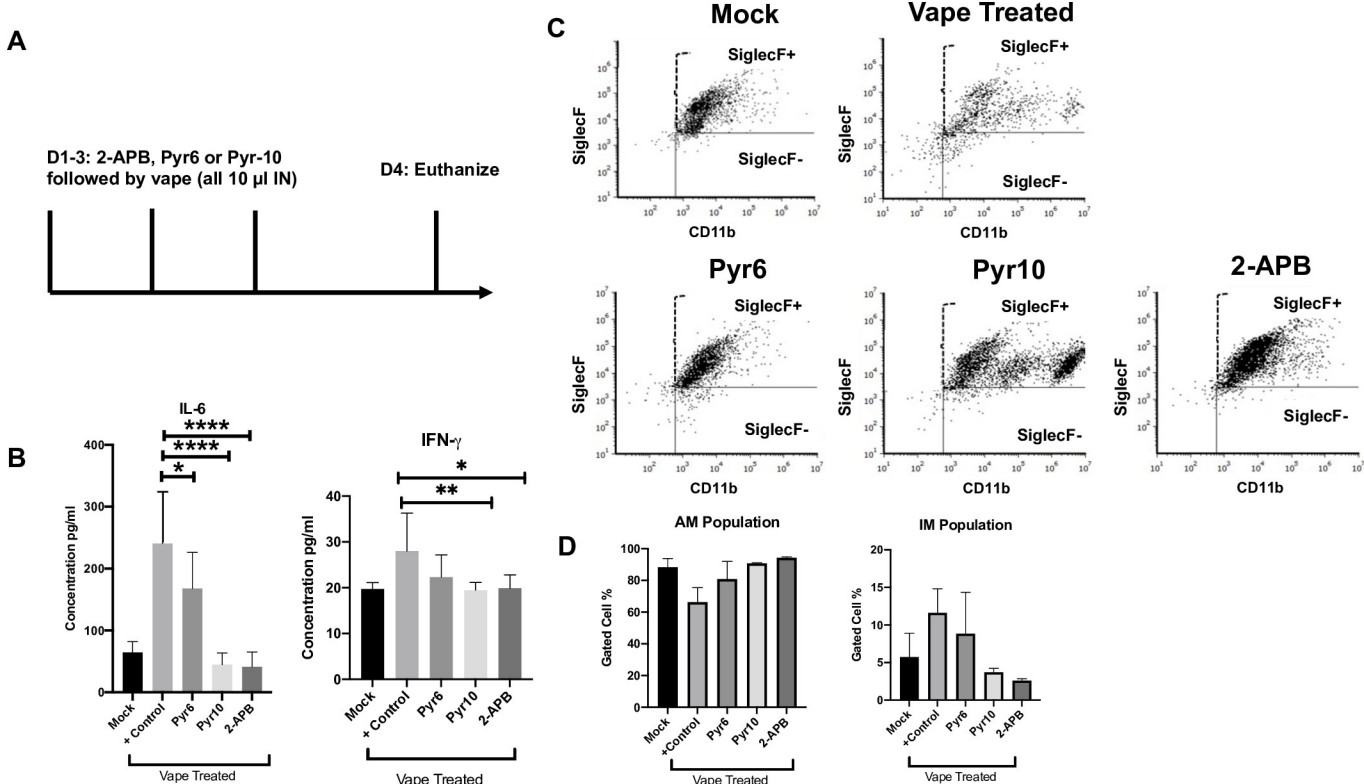

**Fig 2. Increased intracellular Ca²⁺ levels mediate vape-induced inflammation.** (**A**) Study design. Mice were simultaneously vaped and treated with the $Ca^{2+}$ channel antagonists Pyr6, Pyr10 or 2-APB for 3 days. n = 6 mice per treatment group. After being euthanized, BAL fluid was collected, and (**B**) a Luminex cytokine panel including IL-6 and IFN-γ was run using the clarified BAL fluid. (**C**) Cells collected after clarifying the BAL fluid were then analyzed using flow analysis and (**D**) tabulated. n = 2–3 flow analyses per group with a representative image (**C**) for each group shown. IM = SiglecF-/CD11b+ and AM = SiglecF +/CD11b+. Symbols and bars represent the mean ± SEM compared to mock control (* P<0.05, ** P<0.01, **** P<0.0001). AM = alveolar macrophage, IM = interstitial macrophage.

IFN-γ levels were significantly lower for all treated groups, except for Pyr6 which did trend lower, compared to the vape-treated control. We complemented these data through the use of flow cytometry (**Fig 2C and 2D**), where the BAL interstitial macrophage (IM, F4/80+-SiglecF-CD11b+) population of the vape-treated mice was markedly higher than the mock treatment and the 2-APB- and Pyr10-treated groups, which were more similar to the mock-treated group. In contrast, the Pyr-6 treated group was more similar to the vape-treated mice. These findings are intriguing because it is currently well-understood that the majority macrophage population within the airways, and therefore within BAL fluid, is alveolar macrophages (AM, F4/80+SiglecF+CD11b+) and not IM [38].

Based upon these observations, we next asked whether similar results would be observed if the experiments (**Fig 2**) were repeated with the addition of the *K. pneumoniae* challenge on day 4 followed by euthanizing the animals on day 5. As we observed prior vaping to exacerbate *K. pneumoniae*-dependent pathology (**Fig 1**), we evaluated whether this exacerbation was related to Ca²⁺ signaling. Mice were euthanized 24 hpi, and BAL cells and fluid were harvested for analysis (**Fig 3A**). BAL fluid was assessed for inflammatory cytokines, with protein levels found to be generally increased in vaped *vs* mock-treated animals (**Fig 3B**). Interestingly, the three drug treatments produced differential cytokine activation profiles, with Pyr10 treatment correlating to diminished cytokine activation compared to the vaped animals. In general, however, cytokine burdens were decreased in animals administered a Ca²⁺ channel inhibitor, as

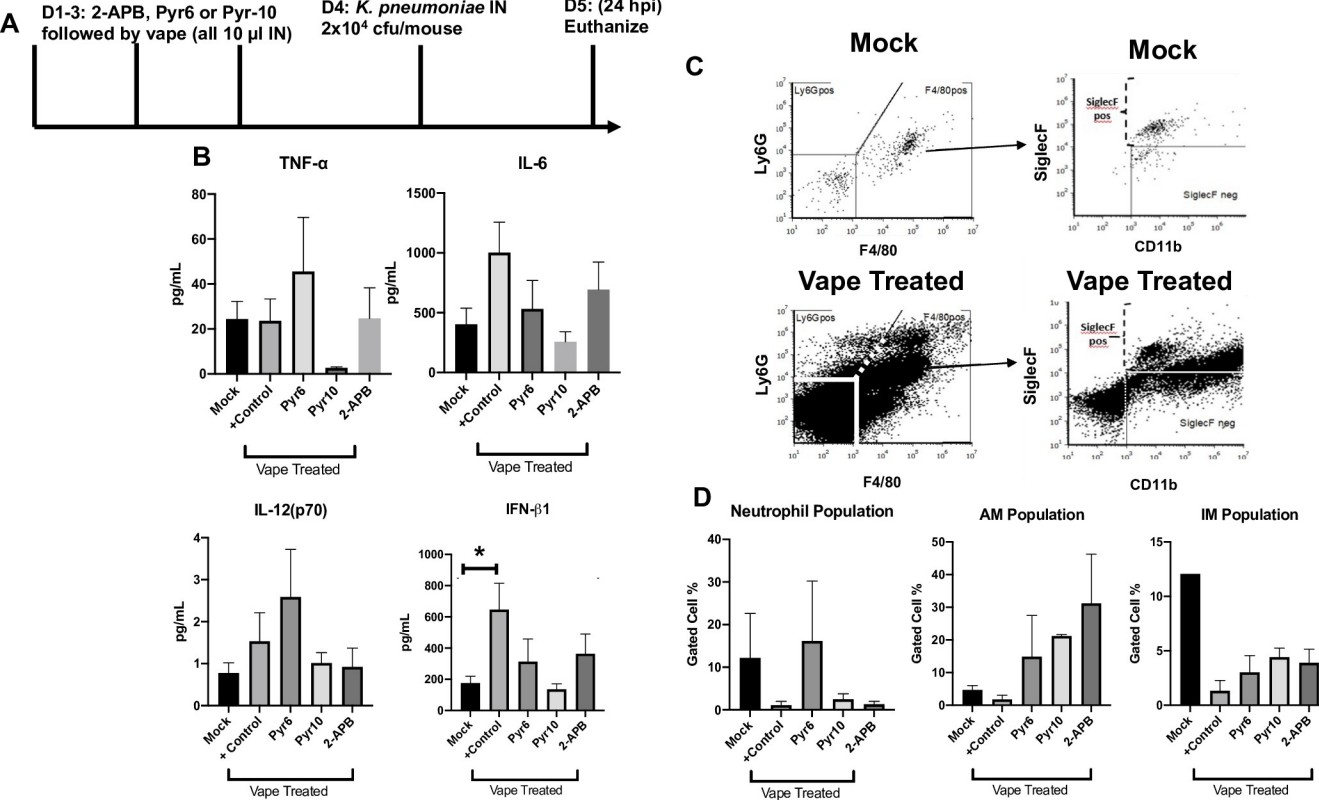

**Fig 3. Ca²⁺ channel antagonists differentially regulate vape-induced inflammation when the treatments are supplemented with a subsequent 24-h pathogen challenge.** (A) Study design. Mice were simultaneously vaped and treated with the $Ca^{2+}$ channel antagonists Pyr6, Pyr10 or 2-APB, then challenged with a sub-lethal dose of *K. pneumoniae* and euthanized for BAL fluid and lung tissue collection. n = 6 mice per treatment group. (B) Clarified BAL was run using the Luminex cytokine panel. (C) Collected BAL fluid cells were then analyzed via flow and (D) tabulated. IM = SiglecF-/CD11b+ and AM = SiglecF +/CD11b+. AM = alveolar macrophage, IM = interstitial macrophage, neutrophils = F4/80-/Lys6G+. Symbols and bars represent the mean ± SEM compared to the mock control (* P<0.05).

compared to the vaped animals (**Fig 3B**). BAL cells were then identified using flow cytometry (**Figs 3C, 3D, S3A and S3B**). Previous studies have indicated that cellular chemotaxis due to *K. pneumoniae* infection is neutrophilic. Therefore, we investigated neutrophil populations within the collected BAL from the vaped animals. Interestingly, neutrophils (F4/80-/Lys6G+) were upregulated in *K. pneumoniae*-infected animals (control group) but not in the vaped animals, except for the Pyr-6 treated group, again, having a different phenotype. In contrast, the cellularity within the BAL from the vaped animals was observed to be primarily monocytes, with an increase in F4/80- / Ly6G- cells. In fact, more IM were observed within the vaped *K. pneumoniae*-infected animals compared to AM, though the percentages were small in comparison to the total population collected. Interestingly, the $Ca^{2+}$ inhibitor drugs appeared to increase the populations of both AM and IM within the lung compared to the vaped only animals.

## Discussion

The inhalation of flavored e-liquid vapors (vaping) has significantly increased in popularity in recent years, yet potential pathologic effects are unclear. Since the introduction of E-cigs, the question of how they impact pulmonary health has been raised. With this study, we were able to further expand upon what is currently known about E-cigs and their role in $Ca^{2+}$

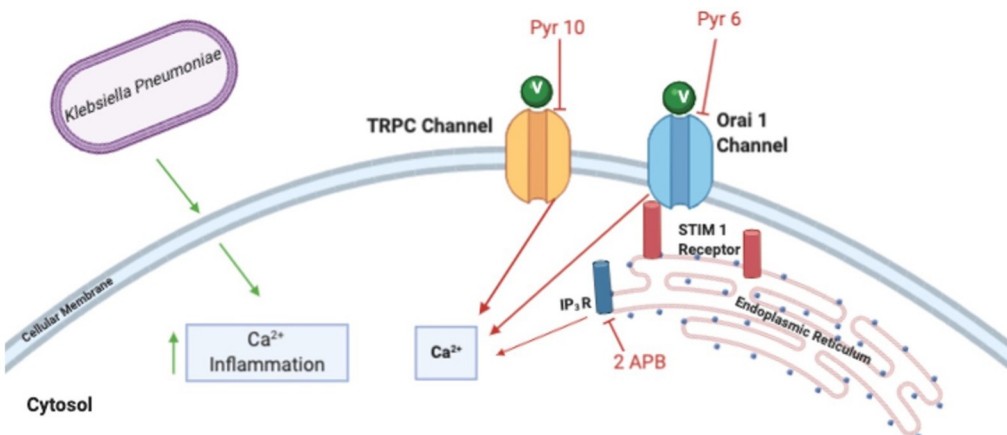

**Fig 4. Schematic representation of the major receptor pathways involved in $Ca^{2+}$ increases due to E-cig/vape exposure.** The $Ca^{2+}$ channel antagonists Pyr6, Pyr10 and 2-APB were utilized to assess their effectiveness in blocking and/or reducing vaping-associated inflammation. Pyr10, Pyr6 and 2-APB inhibit Transient Receptor Protein (TRP) channels, Orai1/STIM1 and the inositol trisphosphate receptor ($IP_3R$), respectively. *K. pneumoniae* infection also elicits an inflammatory response and is also likely to alter $Ca^{2+}$ homeostasis. V = Vape.

mobilization. Using a mouse exposure model, we find that vaping alone leads to mild but observable pulmonary inflammation (**S1 Fig**) [28], which is exacerbated when challenged with a microbial pulmonary infection (**Fig 1**). Interestingly, this inflammation is clearly related to $Ca^{2+}$ mobilization, as pre-treatment with various $Ca^2$ channel antagonists differentially modulated the vaping-induced inflammatory response (**Figs 2, 3**). We found through this vape exposure model that vaped e-liquid potentially interacts with multiple channels, such as TRPC channels, Orai1/STIM1 and $IP_3R$, to impact $Ca^{2+}$ signaling (**Fig 4**). Most of our utilized $Ca^{2+}$ antagonists were able to somewhat reduce inflammatory cytokine production compared to the positive control. Finally, we observed a general decrease in pulmonary inflammation in the vape-treated animal groups pretreated with these same $Ca^{2+}$ antagonists (prior to *K. pneumoniae* infection), with a more marked decrease observed with either Pyr10 or 2-APB treatment (**Fig 3**). However, differential cytokine activation profiles were observable and clearly more work will be required to uncover the role of $Ca^{2+}$ signaling within the context of our dual vape exposure and infection model.

Exposure to traditional tobacco smoke is well-documented to trigger a number of inflammatory responses in the airways and often leads to ailments such as airway inflammation, acute lung injury (ALI), acute respiratory distress syndrome (ARDS) and chronic obstructive pulmonary disease (COPD) [39]. ARDS, in particular, is characterized by acute respiratory failure that results from pulmonary edema and inflammation [40], and its pathophysiology is characterized by increases in circulating levels of multiple inflammatory proteins, including cytokines and chemokines [41, 42]. At the same time, both acute and chronic exposure to cigarette smoke leads to chronic elevations in intracellular $Ca^{2+}$ levels, and smoke from other similar tobacco products, such as E-cigs, is expected to exert similar effects, resulting in pulmonary disease initiation and progression [34, 35, 43]. Although $Ca^{2+}$ has been demonstrated to play a pivotal role in the development and disease progression of ARDS, little is known about the mechanisms of $Ca^{2+}$-induced inflammatory lung diseases.

The recent literature suggests previously underappreciated levels of inflammatory effects and cytotoxicity associated with E-cig use [44–48]. At the same time, vaping renders the user's lungs more susceptible to microbial infection/burden and, therefore, to pulmonary disease [10]. This observation is possibly due to the decreased phagocytic activity of alveolar

macrophages (AMs), which has also been demonstrated with cigarette smoke exposure [49]. However, a functional gap in knowledge does still exist with regard to the effects of vaping upon airway inflammation and microbial infection, which possibly involves macrophage activity.

Our results regarding increased IM populations after vape exposure are intriguing alone (**Fig 2**). However, when factoring in the effects of a simultaneous Pyr6, Pyr10 or 2-APB treatment with the vape exposure, increased $Ca^{2+}$ mobilization becomes a potentially potent mechanism controlling the pro-inflammatory lung response after vape exposure. To our knowledge, there exists a current gap in knowledge surrounding $Ca^{2+}$ signaling (Orai and/or TRP channel signaling (**Fig 4**)) and macrophage activation. While the P2X family of $Ca^{2+}$ channels have been investigated [50], more work must be performed to understand the regulation and function of SOC channels in macrophage phagocytosis and inflammation. Studies that have been performed have predominantly focused on immune activation in the absence of various TRP channels [51, 52] however not on the effects of perturbed $Ca^{2+}$ signaling upon deleterious immune activation, as we observed in our studies. A continuation of our studies (with combined drug exposure or other variation of dosing) will undoubtedly add to this field of study and aid in the development of novel therapeutic interventions for pulmonary diseases.

Our $Ca^{2+}$ channel antagonist exposure model also has potential uses for other respiratory diseases. For example, $Ca^{2+}$ channel blockers are currently employed to treat hypertension and heart arrhythmias [53, 54]. Thus, these $Ca^{2+}$ channel antagonists might find use in the treatment of pulmonary diseases such as COPD, cystic fibrosis or asthma. There is also the potential that $Ca^{2+}$ channel antagonists might inhibit the inflammation observed in EVALI cases.

In sum, we have used an *in vivo* vaping model to demonstrate increased levels of macrophage chemotaxis and inflammatory cytokines after an acute exposure period. Further, we now implicate a $Ca^{2+}$-dependent mechanism leading to this pro-inflammatory response and to immune cell activation. This increase in macrophage activation would likely exacerbate pneumonia symptoms due to pulmonary microbial infection. These findings indicate E-cig use has potential effects on pulmonary health and disease outcome.

## Supporting information

**S1 Fig. An acute *in vivo* model of vaping indicates cytotoxicity.** Mice received either PBS, vaped PG/VG vehicle, resting e-liquid or vaped e-liquid (10 μl) once daily intranasal (IN) for 3 days and were then euthanized. n = 4 mice per treatment group. H&E staining (200X magnification) of sections of lung tissue isolated from mock-, vaped vehicle-, resting e-liquid- or vaped e-liquid- treated mice. Alveolar wall thickening was particularly observable in the lungs from the vaped e-liquid-treated mice compared to the mock control. Arrows indicate alveoli thickening and neutrophil influx.
(PPTX)

**S2 Fig. (A)** Accompanying FACS plot data for Fig 2. **(B)** Multiple FACS scatter plot data frames to accompany Fig 2C. AM = SiglecF+/F4/80+ and IM = SiglecF-/F4/80+.
AM = alveolar macrophage, IM = interstitial macrophage.
(PPTX)

**S3 Fig. (A)** Accompanying FACS plot data for Fig 3. Includes data for all drug treatment groups (2-APB, Pyr-6 and Pyr-10) and *K. pneumoniae* only. **(B)** Multiple FACS scatter plot data frames to accompany Fig 3C. MAC = F4/80+ and NEU = LysG+. MAC = macrophage, NEU = neutrophil.
(PPTX)

## Author Contributions

**Conceptualization:** Rui Zhang, Myles M. Jones, Rob U. Onyenwoke, Vijay Sivaraman.

**Data curation:** De'Jana Parker, Ronna E. Dornsife.

**Formal analysis:** Rui Zhang, Myles M. Jones, Rob U. Onyenwoke.

**Funding acquisition:** Rob U. Onyenwoke.

**Investigation:** Rui Zhang, Myles M. Jones, De'Jana Parker.

**Methodology:** Rui Zhang, Myles M. Jones, De'Jana Parker, Nathan Wymer, Vijay Sivaraman.

**Project administration:** Rob U. Onyenwoke, Vijay Sivaraman.

**Resources:** Rob U. Onyenwoke, Vijay Sivaraman.

**Software:** De'Jana Parker, Ronna E. Dornsife.

**Supervision:** Rob U. Onyenwoke, Vijay Sivaraman.

**Validation:** Vijay Sivaraman.

**Visualization:** Myles M. Jones, De'Jana Parker, Ronna E. Dornsife.

**Writing – original draft:** Rui Zhang, Myles M. Jones.

**Writing – review & editing:** Rob U. Onyenwoke, Vijay Sivaraman.

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
