## [Decision Letter · Decision Letter 0]

23 Jun 2021

PONE-D-21-13261

Acute Vaping Exacerbates Microbial Pneumonia due to Calcium (Ca2+) Dysregulation

PLOS ONE

Dear Dr. Sivaraman,

Thank you for submitting your manuscript to PLOS ONE. After careful consideration, we feel that it has merit but does not fully meet PLOS ONE’s publication criteria as it currently stands. Therefore, we invite you to submit a revised version of the manuscript that addresses the points raised during the review process.

We look forward to receiving your revised manuscript.

Kind regards,

Shama Ahmad, Ph.D.

Academic Editor

PLOS ONE

Journal Requirements:

2. We ask that you please remove citations for unavailable and unpublished work, including manuscripts that have been submitted but not yet accepted (e.g., “unpublished work,” “data not shown”). Instead, include those data as supplementary material or deposit the data in a publicly available database.

3. Thank you for your ethics statement: "This study was performed under authority of the NCCU IACUC. The approved protocol is:

VS-10-09-2019.

For studies, isofluorane was used for anesthesia, and inhaled CO2 was used to euthanasia, as per protocol."

Please amend your current ethics statement to confirm that your named ethics committee specifically approved this study.

For additional information about PLOS ONE submissions requirements for ethics oversight of animal work, please refer to http://journals.plos.org/plosone/s/submission-guidelines#loc-animal-research  

"This work was supported in part by a grant from the National Cancer Institute (NCI) NIH 5-U54-

CA156733-10, National Minority Health Disparities Institute (NIMHD) NIH-3-U54MD012392 and

by funds from NCCU BRITE."

Reviewers' comments:

Reviewer's Responses to Questions

**Comments to the Author**

1. Is the manuscript technically sound, and do the data support the conclusions?

Reviewer #1: Yes

Reviewer #2: Partly

2. Has the statistical analysis been performed appropriately and rigorously? 

Reviewer #1: N/A

Reviewer #2: Yes

3. Have the authors made all data underlying the findings in their manuscript fully available?

Reviewer #1: Yes

Reviewer #2: Yes

4. Is the manuscript presented in an intelligible fashion and written in standard English?

Reviewer #1: Yes

Reviewer #2: Yes

5. Review Comments to the Author

Reviewer #1: This is a very well-written and surprisingly useful mouse study. The information is current, useful and important for interested pulmonary health providers. I have just a few thoughts.

Traditionally the materials and methods are placed before the results, correct?

Vaping resulting in acute cellular lung damage which increases the pathology process in mice infected with K.pneumoniae is important basic information. But why K.pneumoniae? Is it the mouse equivalent of pneumococcal in humans?

Reviewer #2: Zhang et al developed a mouse model of vaped e-liquid pulmonary exposure and studied the effects of calcium ion channel inhibitors in cohorts of mice that received either vaping or vaping+Klebsiella pneumonia. While some of the results are encouraging, the authors overstated their findings – the tested inhibitors barely showed therapeutic effects in vaping+bacterial pneumonia model.

Please proof-read and correct typographical/grammar issues.

Abstract:

The authors wrote – “Using this model, we demonstrate through the use of clinical measures, that is, the lung wet:dry ratio and lung histology/H&E staining, lung pathology.” Although these parameters are clinically used but only in EVALI-related dead patients or post-mortem. These parameters are widely used in pre-clinical models. Lung histology and lung pathology can be synonymously used, therefore one of these two words can be deleted.

The authors wrote – “This increase in macrophage activation appears to exacerbate pneumonia when due to pulmonary microbial infection.” Not clear, please paraphrase.

Results:

The authors wrote “To that end, we developed a protocol utilizing commercially available E-cig devices to aerosolize the e-liquid and then condensed the vapors back into a “vaped” e-liquid. With this new reagent in hand, we first assessed its toxicity in vitro (data not shown, manuscript under review).” In my opinion, the manuscript under review should be removed.

Page 6 – “lung pathology of the K. pneumoniae-infected only animals also demonstrated a mild inflammatory influx of neutrophils within the alveoli, though the majority of airways were unimpeded (Fig. 1B).” It should be Fig. 1C.

In figure 1B, on the y-axis, lung weight (mg) is written. Ratios have no units.

In figure 1C, instead of pixilation, please use scoring methods such as ATS scoring for ALI An official American Thoracic Society workshop report: features and measurements of experimental acute lung injury in animals - PubMed (nih.gov) (PMID: 21531958).

In Figures 2A and 2B, please use consistent labeling for axes. Present vape+control bar after sham (control) group for easy interpretation here and subsequent bar graphs. If possible, please present bar graphs with scatter plots to see the variation of the data.

For ease of interpretation, please present the study paradigm (as in Figure 1A) in a panel for figures 2 and 3. By looking at legends for figures 2 and 3, it is difficult to differentiate the study paradigm.

6. PLOS authors have the option to publish the peer review history of their article (what does this mean?). If published, this will include your full peer review and any attached files.

Reviewer #1: No

Reviewer #2: No

---

## [Author Response · Author response to Decision Letter 0]

13 Jul 2021

2. We ask that you please remove citations for unavailable and unpublished work, including manuscripts that have been submitted but not yet accepted (e.g., “unpublished work,” “data not shown”). 

 We have removed all “data not shown” and replaced with a citation for our recent publication. Lines 209 and 323.

Please amend your current ethics statement to confirm that your named ethics committee specifically approved this study.

 All animal care and use were conducted in accordance with the Guide for the Care and Use of the Laboratory Animals (National Institutes of Health). The study was carried out in strict accordance with the NCCU IACUC (protocol number VS-10-09-2019). Mice were maintained at 25°C and 15% relative humidity with alternating 12-h light /dark periods. All exposures and surgeries were performed under isofluorane anesthesia, and all efforts were made to minimize suffering.

"This work was supported in part by a grant from the National Cancer Institute (NCI) NIH 5-U54-

CA156733-10, National Minority Health Disparities Institute (NIMHD) NIH-3-U54MD012392 and

by funds from NCCU BRITE."

We apologize for this mistake. This has been updated in the cover page. Please include the following text for our funding statement:

“This work was supported in part by a grant from the National Cancer Institute (NCI) NIH 5-U54-CA156733-10, National Minority Health Disparities Institute (NIMHD) NIH-3-U54MD012392 and by funds from NCCU BRITE.”

5. We note that you have included the phrase “data not shown” in your manuscript. 

 We have removed all “data not shown” and replaced with a citation for our recent publication. Lines 209 and 323.

Reviewers' comments:

Traditionally the materials and methods are placed before the results, correct? ---------Now corrected.

Vaping resulting in acute cellular lung damage which increases the pathology process in mice infected with K.pneumoniae is important basic information. But why K.pneumoniae? Is it the mouse equivalent of pneumococcal in humans? ----------Excellent point by the reviewer. We now include text for our rationale of using K. pneumoniae. Lines 215-217.

Reviewer #2: Zhang et al developed a mouse model of vaped e-liquid pulmonary exposure and studied the effects of calcium ion channel inhibitors in cohorts of mice that received either vaping or vaping+Klebsiella pneumonia. While some of the results are encouraging, the authors overstated their findings – the tested inhibitors barely showed therapeutic effects in vaping+bacterial pneumonia model.

-------------We have now amended the text to indicate the obtained data are promising but not complete and that further study will be required to understand this mechanism (Line 329-335).

Abstract:

The authors wrote – “Using this model, we demonstrate through the use of clinical measures, that is, the lung wet:dry ratio and lung histology/H&E staining, lung pathology.” Although these parameters are clinically used but only in EVALI-related dead patients or post-mortem. These parameters are widely used in pre-clinical models. Lung histology and lung pathology can be synonymously used, therefore one of these two words can be deleted.----------Edited and now done. Lines 50-52.

The authors wrote – “This increase in macrophage activation appears to exacerbate pneumonia when due to pulmonary microbial infection.” Not clear, please paraphrase. ---------Edited. Lines 54-56.

Results:

The authors wrote “To that end, we developed a protocol utilizing commercially available E-cig devices to aerosolize the e-liquid and then condensed the vapors back into a “vaped” e-liquid. With this new reagent in hand, we first assessed its toxicity in vitro (data not shown, manuscript under review).” In my opinion, the manuscript under review should be removed. -----------Removed and replaced. We now reference our recent publication (Sivaraman et al., Front. Physiol., 2021) for these data, which are now freely available. Lines 208-209.

Page 6 – “lung pathology of the K. pneumoniae-infected only animals also demonstrated a mild inflammatory influx of neutrophils within the alveoli, though the majority of airways were unimpeded (Fig. 1B).” It should be Fig. 1C.--------- Done. Line 228.

In figure 1B, on the y-axis, lung weight (mg) is written. Ratios have no units. --------Done.

In figure 1C, instead of pixilation, please use scoring methods such as ATS scoring for ALI An official American Thoracic Society workshop report: features and measurements of experimental acute lung injury in animals - PubMed (nih.gov) (PMID: 21531958).---------- ATS scoring for ALI has now been performed and is now included within Figure 1C. Please see also lines 187-193 and 243-244.

In Figures 2A and 2B, please use consistent labeling for axes. Present vape+control bar after sham (control) group for easy interpretation here and subsequent bar graphs. ---------Done.

If possible, please present bar graphs with scatter plots to see the variation of the data. ---------This is an excellent point. These data sets are, however, very large and were collected and analyzed by the NC Central University Flow Core. We did ask for and receive additional raw data scatter plots from the Core and now include these plots to allow for readers to observe the variation of the data (Supplementary Figures 2B and 3B). The quantitative data are currently within Figures 2 and 3.

For ease of interpretation, please present the study paradigm (as in Figure 1A) in a panel for figures 2 and 3. By looking at legends for figures 2 and 3, it is difficult to differentiate the study paradigm.----------- Study paradigms are now included for Figures 2 and 3, i.e., Figures 2A and 3A.

---

## [Decision Letter · Decision Letter 1]

2 Aug 2021

Acute Vaping Exacerbates Microbial Pneumonia due to Calcium (Ca2+) Dysregulation

PONE-D-21-13261R1

Dear Dr. Sivaraman,

We’re pleased to inform you that your manuscript has been judged scientifically suitable for publication and will be formally accepted for publication once it meets all outstanding technical requirements.

Kind regards,

Shama Ahmad, Ph.D.

Academic Editor

PLOS ONE

Additional Editor Comments (optional):

Reviewers' comments:

Reviewer's Responses to Questions

**Comments to the Author**

1. If the authors have adequately addressed your comments raised in a previous round of review and you feel that this manuscript is now acceptable for publication, you may indicate that here to bypass the “Comments to the Author” section, enter your conflict of interest statement in the “Confidential to Editor” section, and submit your "Accept" recommendation.

Reviewer #1: All comments have been addressed

Reviewer #2: All comments have been addressed

2. Is the manuscript technically sound, and do the data support the conclusions?

Reviewer #1: Yes

Reviewer #2: Yes

3. Has the statistical analysis been performed appropriately and rigorously? 

Reviewer #1: N/A

Reviewer #2: Yes

4. Have the authors made all data underlying the findings in their manuscript fully available?

Reviewer #1: Yes

Reviewer #2: Yes

5. Is the manuscript presented in an intelligible fashion and written in standard English?

Reviewer #1: Yes

Reviewer #2: Yes

6. Review Comments to the Author

Reviewer #1: Well-done, interesting and educational paper. The author addressed and answered all of the reviewers' observations and questions adequately and completely. The selection of the specific microorganism was explained in a complete and referenced sentence. This is overall worth reading for primary care and specialty care providers and, in my opinion, should be accepted and published. While I am thinking about this, I have decided that I don't like to work with a 'minimum character count' and since I am volunteering my effort, I will ask that you remove my name from your list of reviewers.

Reviewer #2: (No Response)

7. PLOS authors have the option to publish the peer review history of their article (what does this mean?). If published, this will include your full peer review and any attached files.

Reviewer #1: No

Reviewer #2: No

---

## [Editor Report · Acceptance letter]

4 Aug 2021

PONE-D-21-13261R1 

Acute Vaping Exacerbates Microbial Pneumonia due to Calcium (Ca^2+^) Dysregulation 

Dear Dr. Sivaraman:

I'm pleased to inform you that your manuscript has been deemed suitable for publication in PLOS ONE. Congratulations! Your manuscript is now with our production department. 

Kind regards, 

on behalf of

Dr. Shama Ahmad 

Academic Editor

PLOS ONE